# What Types of Exercise Are Best for Emotional Intelligence and Logical Thinking?

**DOI:** 10.3390/ijerph191610076

**Published:** 2022-08-15

**Authors:** Albertas Skurvydas, Ausra Lisinskiene, Daiva Majauskiene, Dovile Valanciene, Ruta Dadeliene, Natalja Istomina, Asta Sarkauskiene, Gediminas Buciunas

**Affiliations:** 1Education Academy, Vytautas Magnus University, K. Donelaičio Str. 58, 44248 Kaunas, Lithuania; 2Department of Rehabilitation, Physical and Sports Medicine, Institute of Health Sciences, Faculty of Medicine, Vilnius University, 21/27 M.K. Čiurlionio St., 03101 Vilnius, Lithuania; 3Institute of Education, Studies, Education Academy, Vytautas Magnus University, K. Donelaičio Str. 58, 44248 Kaunas, Lithuania; 4Institute of Health Sciences, Faculty of Medicine, Vilnius University, 21/27 M.K. Čiurlionio St., 03101 Vilnius, Lithuania; 5Departments of Sports Recreation and Tourism, Klaipėda University, Herkaus Manto St. 84, 92294 Klaipėda, Lithuania; 6Faculty of Law, Vytautas Magnus University, K. Donelaičio Str. 58, 44248 Kaunas, Lithuania

**Keywords:** emotional intelligence, logical thinking, physical activity, age, sex, exercise specifics

## Abstract

The aim of our study was to determine whether EI and LT vs. intuitive thinking (CRT score) are related to participation in professional sports, independent exercise, and exercise at a gym/health center compared with no exercise. We selected 20 of the most popular types of exercise in Lithuania among respondents who exercise independently or at a gym/health center, and we ranked these types of exercise according to the participants’ emotional intelligence and logical thinking. We studied 4545 women and 1824 men aged 18–74 years with a focus on whether emotional intelligence and logical thinking are related to type of exercise. Participation in any exercise was significantly related to emotional intelligence in men and women. Women in professional sports solved the lowest number of logic tasks. Women who exercise independently or at a gym/health center had better logical thinking than those who do not exercise. Among men, logical thinking was not associated with the type of exercise. We found the tendency for a negative correlation between EI and LT in the 20 most popular types of exercise. Emotional intelligence correlated positively with participation in MVPA. The highest emotional intelligence was in women who participate in dance or Pilates and in men who participate in martial arts, wrestling, boxing, or yoga. Logical thinking was the highest in men who participate in triathlon and in women who perform CrossFit. Men who practice martial arts or track and field and women who participate in cycling were in the top five for emotional intelligence and logical thinking.

## 1. Introduction

There is an increasing amount of evidence-based research showing that various forms and doses of physical activity (PA) or exercise are effective in combating many chronic diseases [1,2] and improving well-being and mental health [3,4,5,6,7,8,9]. The health benefits of PA are influenced by age, sex, health, and body mass index (BMI) [10,11,12,13]. Studies have also shown that the effects of PA or exercise on the body are specific; that is, they are affected by the intensity, duration, type of muscle contraction, motor control, and other factors [14,15]. Other findings suggest that PA or exercise may help in the maintenance of, or prevention of the decline in, cognitive function with illness or aging [16,17,18]. Insufficient PA is related to obesity and, through obesity, may contribute to systemic inflammation throughout the body, which is causally related to several chronic diseases [1,19].

Researchers have emphasized the distinction between two types of cognitive processes: those executed quickly (“fast” thinking), involving little conscious deliberation, and those that are slower and more reflective (“slow”, analytical thinking (LT)) [20]. Dual-process models propose that health actions are guided by both a conscious, reflective, analytical, rule-based system and a nonconscious, impulsive, associative system [20,21]. Explicit thinking (i.e., LT) requires more effort than implicit (i.e., “fast”) thinking, and implicit thinking is more often used to make rapid decisions [20,21]. 

Some studies investigated how different types of PA and exercise affect various indicators of human health. Berger and Owen established the positive effect of four exercise regimes (swimming, body conditioning, hatha yoga, fencing) on reducing stress and improving mood [22]. Matias et al. studied how depression depends on PA type (walking, running, cycling, team sports) and established reduced depressive symptoms [18]. Wennman and Borodulin investigated the association between PA type (e.g., walking, cleaning, jogging, swimming, skiing, gardening, stair climbing, and others) and meeting the PA guidelines [23]. Chekroud et al. studied the association between PA (aerobic or gym exercise, cycling, housework, popular sports, recreation, running or jogging, walking, and winter and water sports) and mental health in 1.2 million individuals and found that engaging in team sports and cycling was associated with the lowest mental health burden [24]. Schnohr et al. reported that the gains in life expectancy were larger in those who participate in specific sports compared with sedentary groups; the gains in life expectancy associated with specific activities were as follows: tennis, 9.7 years; badminton, 6.2 years; soccer, 4.7 years; cycling, 3.7 years; swimming, 3.4 years; jogging, 3.2 years; calisthenics, 3.1 years; and health club activities, 1.5 years [25]. It was, therefore, concluded that the type of activity may determine the magnitude of the improvement in life expectancy. However, the study by Schnohr et al. involved an observational study, and it remains uncertain whether this relationship is causal. Interestingly, the leisure-time sports that inherently involve more social interaction were associated with the best longevity, a finding that warrants further investigation [25].

Few researchers have examined whether EI and LT relate to the exercise or PA level that people perform to enhance health and during active leisure time. The aim of our study was to determine whether EI and LT vs. intuitive thinking (CRT score; [26]) are related to participation in professional sports, independent exercise, and exercise at a gym/health center compared with no exercise. We ranked the 20 most popular types of exercise among people who are not professional athletes in Lithuania to determine whether the type of independent PA or exercise is related to EI and LT in men and women of different ages.

## 2. Materials and Methods

### 2.1. Participants

We surveyed 6369 people. The participants were aged 18–74 years (Table 1). The study was conducted from October 2019 to June 2020. The participants were selected from Lithuania to represent a Lithuanian sample. Participation was anonymous, and data collection and processing were confidential. We used an online survey to collect information using Google Forms (https://docs.google.com/forms/) (accessed on 1 March 2020). All participants completed the online questionnaires. The online survey was distributed by researchers through social media (Facebook) and personal messaging (WhatsApp). From the survey, we determined the BMI (kg/m^2^) and specificity of exercise of the participants, i.e., what percentage of the participants did not perform any physical exercise, were professional athletes, performed physical exercises independently, and exercised in sports and health centers (Table 1). The most popular sports among professional athletes were: martial arts (*n* = 34; both men and women), track and field (*n* = 30), dance (*n* = 25), volleyball (*n* = 24), basketball (*n* = 24), fitness (*n* = 19), running (*n* = 18), football (*n* = 17), roller sports (*n* = 14), powerlifting (*n* = 12), wrestling (*n* = 12), tennis (*n* = 10), handball (*n* = 9), boxing (*n* = 9), swimming (*n* = 9), rowing (*n* = 6), CrossFit (*n* = 6), aerobics (*n* = 6), bodybuilding (*n* = 5), triathlon (*n* = 5), and cycling (*n* = 3). As the number of participants in each of the sports was not large, we did not examine them separately but combined them into one group—professional athletes. From those who exercised independently and those who exercised at gyms/health centers, we selected 20 of the most popular exercise types among men and women in Lithuania in total, 2324 women and 1176 men were in the “20” group (individually exercise nonprofessionally or at gyms/health centers, performing one/some of the 20 most popular types of exercise in Lithuania).

### 2.2. Measurements

Danish Physical Activity Questionnaire (DPAQ): the DPAQ was adapted from the International Physical Activity Questionnaire and differs from it by referring to PA in the past 24 h (for 7 consecutive days) instead of the past 7 days. The selected activities were listed on the PA scale at nine levels of physical exertion in metabolic equivalents (METs), ranging from sleep or inactivity (0.9 MET) to highly strenuous activities (>6 METs). Each level (A = 0.9 MET, B = 1.0 MET, C = 1.5 METs, D = 2.0 METs, E = 3.0 METs, F = 4.0 METs, G = 5.0 METs, H = 6.0 METs, and I > 6 METs) is described using examples of specific activities of that particular MET level and a small drawing. The PA scale is constructed so that the number of minutes (15, 30, or 45) and hours (1–10) spent at each MET activity level on an average 24 h weekday can be filled out. This allows for a calculation of the total MET time, representing 24 h of sleep, work, and leisure time on an average weekday [27]. We calculated how much energy (in METs) was consumed per day during sleep, moderate-intensity PA (MPA; 3–6 METs), and vigorous-intensity PA (VPA; >6 METs). We also combined MPA with VPA as moderate-to-vigorous PA (MVPA). 

The Cognitive Reflection Test (CRT): the test tasks were developed according to the CRT test discussed in the article by [26]. The test consists of three tasks, which, after reading, automatically select the wrong answer. The author noted that it is possible to check what kind of thinking system a person uses. The first system reflects intuitive decision-making, which is usually fast, automatic, implicit, emotional, and requires minimal effort. Meanwhile, the second system reflects reasoning that is slower, conscious, effort intensive, goal oriented, and logical. The test consists of three questions, for example: (1) A bat and a ball cost $1.10 in total. The bat costs $1.00 more than the ball. How much does the ball cost? _____ cents; (2) If it takes 5 machines 5 min to make 5 widgets, how long would it take 100 machines to make 100 widgets? _____ minutes; (3) In a lake, there is a patch of lily pads. Every day, the patch doubles in size. If it takes 48 days for the patch to cover the entire lake, how long would it take for the patch to cover half of the lake? _____ days. The measure is scored as the total number of correct answers. The Cognitive Reflection Test (CRT) measures cognitive processing—specifically, the tendency to suppress an incorrect, intuitive answer and come to a more deliberate, correct answer via LT.

Emotional Intelligence (EI): EI was assessed using the Schutte self-report EI test (SSREIT) [28]. The SSREIT is a 33-item questionnaire divided into four subscales: perception of emotion assessed by 10 items, managing own emotions assessed by 9 items, managing others’ emotions assessed by 8 items, and utilization of emotions assessed by 5 items. The items are answered on a five-point scale ranging from 1 (strongly disagree) to 5 (strongly agree). Total scores range from 33 to 165, with the higher scores indicating greater ability in EI.

### 2.3. Statistical Analysis

Interval data are reported as the mean ± standard error. All interval data were confirmed as being normally distributed using the Kolmogorov–Smirnov test. Three-way analyses of variance (ANOVA) were performed to assess the effect of independent variables (specificity of exercise, age, gender) on the dependent (CRT score and EI) variables. Two-way analyses of variance (ANOVA) were performed to assess the effect of independent variables (specificity of exercise and age) on the dependent (CRT score and EI) variables in females and males. The partial eta-squared (${ŋ}_{P}^{2}$) value was estimated as a measure of the effect size. If significant effects were found, Tukey’s post hoc adjustment was used for multiple comparisons within each repeated ANOVA. We also calculated the Pearson coefficient of correlation. Moreover, we computed the chi-square (${ŋ}_{P}^{2}$) and its *p*-value. For all tests, statistical significance was defined as *p* < 0.05. Statistical analyses were performed using IBM SPSS Statistics software (version 22; IBM SPSS, Armonk, NY, USA).

## 3. Results

### 3.1. Descriptive Values

The number, age, exercise experience, and number of sports for the men and women from all study groups are presented in Table 1. Study groups differed in age (*p* < 0.001; ${ŋ}_{P}^{2}$ = 0.024) and sex distribution (*p* < 0.001; ${ŋ}_{P}^{2}$ = 0.003), but the interaction between age and sex was not significant. Professional athletes were younger than the other groups (*p* < 0.001). Male and female professional athletes participated in exercise or PA for longer than those who exercise independently or at a gym/health center (*p* < 0.001). Those who exercise independently had greater exercise experience than those who exercise at a gym/health center (*p* < 0.001). The duration of exercise experience was longer in all groups of men (*p* < 0.001). BMI was lower in men and women in professional sports than in the other groups and lower in men and women who exercise independently or who exercise at a gym/health center than in those who do not exercise.

**Table 1 ijerph-19-10076-t001:** Number, age, and exercise experience of the study participants.

		I Don’t Exercise		I’m in a Professional Sport		I Exercise by Myself		I Exercise in a Gym/Health Center		Associated with of Exercise
Count %	Women	1726	38.00%	154	3.40%	1327	29.20%	1338	29.4%	
	Men	394	21.60%	139	7.60%	875	48%	416	22.8%	
Age, years	Women	39 _b_	11.8	29.1 _a_	10.5	39 _b_	11.3	37.9	12.6	<0.001
	Men	37.7 _b_	11.1	28.7 _a_	9.3	35.7 _c_	10	35.8 _c_	11.4	<0.001
Exercise experience	Women	0		9.5 _a_	7.7	6.3 _b_	8.6	5.2	7.1	<0.001
	Men	0		12 _a_	8.1	10.8 _b_	11.1	7.8	8.8	<0.001
Total types of exercises		0		44		66		74		<0.001

Letters that do not match indicate *p* < 0.05.

All indicators of EI (except for use of emotion for men) were worse for those who do not exercise than in the other groups (Table 2). Logical thinking (LT) in men was not significantly related to the type of exercise. By contrast, in women, the lowest LT score was in professional athletes (*p* < 0.05 compared with other groups), and the highest was for those who exercise independently or at a gym/health center. The MVPA was higher in male and female professional athletes than in those in the other groups (*p* < 0.001) and higher in men and women who exercise independently and those who exercise at a gym/health center than in untrained participants (who do not exercise). Sleep METs was higher in male and female professional athletes than in the other groups; men who do not exercise had the lowest sleep METs (*p* < 0.05).

The characteristics of the people who participate in the 20 most popular types of exercise in Lithuania are presented in Table 3. The type of exercise was significantly related to BMI in both women and men (women: *p* < 0.001, ${ŋ}_{P}^{2}$ = 0.048; men: *p* < 0.001, ${ŋ}_{P}^{2}$ = 0.06), MVPA (women: *p* < 0.001, ${ŋ}_{P}^{2}$ = 0.152; men: *p* < 0.001, ${ŋ}_{P}^{2}$ = 0.132), and sleep (women: *p* = 0.13, ${ŋ}_{P}^{2}$ = 0.01; men: *p* < 0.001, ${ŋ}_{P}^{2}$ = 0.041) (Table 3).

**Table 2 ijerph-19-10076-t002:** Moderate-to-vigorous physical activity, sleep, body mass index, emotional intelligence, and logical thinking in women and men.

		I Don’t Exercise		I’m in a Professional Sport		I Exercise by Myself		I Exercise in a Gym/Health Center		Associated with of Exercise
Moderate-to-	Women	7.78 _a_	7.4	23.4 _b_	12.9	14.7 _c_	10.6	14.8 _c_	8.9	<0.001
vigorous PA, METs	Men	9.3 _a_	11.2	25 _b_	11.2	17.6 _c_	11.8	17.1 _c_	10.6	<0.001
Sleep, METs	Women	6.55	1	6.84 _a_	0.86	6.6	0.9	6.57	0.78	0.92
	Men	6.33 _a_	0.94	6.86 _b_	0.91	6.53 _c_	0.92	6.52 _c_	0.79	<0.001
Body mass index, kg/m^2^	Women	25.2 _a_	5.2	22.3 _b_	3.1	23.5 _c_	3.8	23.4 _c_	4	<0.001
	Man	26.7 _a_	4.1	24.8 _b_	3.5	26 _c_	3.3	25.6 _c_	3.5	<0.001
Emotional intelligence	Women	124.8 _a_	16.8	131.1	13.2	128.9	15.4	128.6	15.4	<0.001
	Men	119.8 _a_	16.5	126.9	14.1	123.7	15.1	127.5	14.9	<0.001
Perception of emotion	Women	37.1 _a_	6.1	39.2	5.1	38.3	5.3	38.2	5.7	<0.001
	Men	35.4 _a_	5.9	37.9	5.6	36.8	5.6	36.9	5.9	<0.001
Managing own emotion	Women	34.7 _a_	5.4	36.9	4.1	36.1	5	36	5	<0.001
	Man	34.3 _a_	5.3	36.3	4.6	35.4	4.9	35.3	4.7	<0.001
Managing others emotion	Women	29.8 _a_	4.6	31	4.1	30.7	4.4	30.7	4.3	<0.001
	Men	28 _a_	4.7	29.9	4.5	28.9	4.5	28.9	4.5	<0.001
Utilization	Women	22.7 _a_	3.9	23.9	3.6	23.9	3.7	23.8	3.7	<0.001
emotion	Men	22.1	4.2	22.9	3.5	22.6	3.7	22.7	3.7	0.09
Logical	Women	1.02 _a_	1.1	0.86 _b_	1	1.09 _c_	1.1	1.14 _c_	0.9	0.003
thinking	Men	1.24	1.1	1.31	1.1	1.25	1.1	1.23	1.1	0.9

Letters that do not match indicate *p* < 0.05.

### 3.2. Associated with of Exercise Specifics on the EI and LT of Men and Women of Different Ages

EI was lower in men and women of all ages who do not exercise than in the other groups (*p* < 0.001), but EI did not differ significantly (*p* > 0.05) between the other groups (Figure 1). EI was higher in women than in men (*p* < 0.001). In addition, EI differed between age groups (*p* < 0.007), and the interaction between the type of exercise and age was significant (*p* < 0.035). All logic tasks were solved by 16.5% of women who do not exercise, 13.6% of female professional athletes, 18.0% of women who exercise independently, and 19.4% of women who exercise at a gym/health center (chi-square: 19.7; *p* = 0.02). The respective values for men were 20.3%, 25.2%, 23.5%, and 20.0%, (chi-square: 8.7; *p* = 0.46). Female professional athletes aged 18–25 and 26–44 years had the lowest logic task scores compared with other groups of the same age (*p* < 0.01). A three-way ANOVA showed that solving logic tasks was significantly related to age (*p* = 0.045) and sex (*p* = 0.003) but not to the type of exercise (*p* = 0.297); the interaction between the three factors was not significant.

### 3.3. Ranking of 20 Types of Exercise According to EI

The associated with of the type of exercise (20 types) on EI was not significant in women and men: women: *p* = 0.291, ${ŋ}_{P}^{2}$ = 0.013; men: *p* = 0.507, ${ŋ}_{P}^{2}$ = 0.019 (two-way ANOVA) (Table 4). However, associated with of age on EI was significant in women (*p* = 0.002) and men (*p* = 0.001). The general EI was higher in women who participate in dance and Pilates than in those who perform CrossFit (*p* < 0.05, post hoc Tukey analysis); there were no significant differences between the other exercise types. The only significant difference among men was between those who participate in martial arts, wrestling, boxing, or yoga and those who perform powerlifting. In both men and women, the type of exercise was not significantly related to EI scores for perception (male: *p* = 0.523, ${ŋ}_{P}^{2}$ = 0.019; female: *p* = 0.63, ${ŋ}_{P}^{2}$ = 0.008), managing own emotions (male: *p* = 0.57, ${ŋ}_{P}^{2}$ = 0.018; female: *p* = 0.206, ${ŋ}_{P}^{2}$ = 0.008), managing others’ emotions (female: *p* = 0.115, ${ŋ}_{P}^{2}$ = 0.013; male: *p* = 0.163, ${ŋ}_{P}^{2}$ = 0.011), and use of emotions (male: *p* = 0.143, ${ŋ}_{P}^{2}$ = 0.019; female: *p* = 0.817, ${ŋ}_{P}^{2}$ = 0.005) (Table 5). All EI scores (except use of emotions) were significantly dependent on age (associated with of age: *p* < 0.01). In all cases, the interaction between the two factors was not significant.

Among women, scores for EI perception, managing own emotions, and managing others’ emotions were highest among those who participate in dance and Pilates. Scores for managing own and others’ emotions were also highest among women who play volleyball, and EI scores were significantly higher in those who participate in CrossFit.

Among men, those who participate in wrestling and boxing had the highest EI perception score, which was significantly higher than in those who perform powerlifting. Men who participate in track and field and martial arts had the highest scores for managing their own emotions, which were significantly higher than in those who exercise at a gym/health center. Men who participate in martial arts or boxing had the highest scores for managing others’ emotions, which were higher than in those who perform powerlifting. Men who participate in yoga had the highest use of emotion scores, which were significantly higher than in those who perform powerlifting.

**Table 4 ijerph-19-10076-t004:** General emotional intelligence in men and women according to the type of exercise.

Men			Women		
Sport	Emotional Intelligence General Rank		Sport	Emotional Intelligence General Rank	
	Mean	SD		Mean	SD
Martial arts	127.84	14.29	Dance	132.5	15.02
Wrestling	127.71	13.66	Pilates	132.2	12.68
Boxing	127.08	15.07	Volleyball	131.4	16.54
Yoga	127.06	10.7	Cycling	130.3	15.07
Track and field	125.73	12.23	Track and field	130.2	15.80
CrossFit	125.41	13.67	Fitness	130.0	14.70
Running	125.14	14.07	Tennis	129.7	13.90
Independent exercise	124.72	16.3	Swimming	129.7	18.29
Volleyball	124.5	14.32	Martial arts	129.4	13.24
Football	124.31	12.77	Basketball	129.2	14.40
Basketball	124.19	14.76	Stretching	129.1	16.56
Bodybuilding	124.18	14.5	Yoga	129.1	15.92
Fitness	124.14	17.91	Aerobics	129.1	14.84
Swimming	122.83	15.52	Gym	129.1	19.51
Resistance exercise	122.48	15.68	Resistance exercise	128.8	14.93
Triathlon	121.65	16.05	HIT	128.7	13.20
Cycling	121.05	15.36	Running	128.2	14.81
Tennis	120.31	14.01	Independent exercise	127.9	15.56
Gym	119.78	17.98	Callanetics	126.8	15.14
Powerlifting	119.27	13.57	CrossFit	123.6	13.83
Untrained	120.16	16.84	Untrained	125.1	16.81
Professional athletes	126.9	14.5	Professional athletes	131.1	13.20

**Table 5 ijerph-19-10076-t005:** Relationship between different components of emotional intelligence and type of exercise in men and women.

**Men**								
**Sport**	**Perception of Emotion**		**Managing Own Emotion**		**Managing Others’ Emotion**		**Utilization Emotion**	
	**Mean**	**SD**	**Mean**	**SD**	**Mean**	**SD**	**Mean**	**SD**
Basketball	37.22	5.53	35.55	4.72	29.02	4.38	22.4	3.56
Bodybuilding	36.21	5.04	35.34	5.48	29.58	4.05	23.05	4.08
Boxing	38.32	5.82	36.12	4.82	29.64	4.36	23	4.05
CrossFit	37.71	5.1	35.71	4.13	29.4	4.52	22.6	3.9
Cycling	35.66	6.25	35.25	4.56	28.21	4.81	21.95	3.9
Fitness	36.52	6.45	35.05	5.5	29.6	5.25	22.98	4.48
Football	37.23	4.93	35.25	4.53	28.69	3.63	23.15	3.3
Gym	35.83	6.9	34.28	5.38	27.67	3.94	22	3.96
Independent exercise	36.65	6.29	35.19	5.27	29.54	4.88	23.34	3.74
Martial arts	38.2	5.95	36.43	4.33	29.76	4.35	23.46	3.61
Powerlifting	35.6	6.16	35.93	4.68	26.73	4.53	21	2.67
Resistance exercise	36.83	6.04	35.1	4.88	28.07	4.55	22.48	3.76
Running	36.87	5.37	35.95	4.71	29.41	4.25	22.91	3.83
Swimming	37.76	4.87	34.29	5.05	28.22	4.89	22.56	3.19
Tennis	36.83	5.63	34.63	4.81	27.46	4.05	21.4	3.99
Track and field	36.77	4.53	36.81	4.51	29.46	4.83	22.69	3.11
Triathlon	35.6	4.48	35.05	4.91	28.8	4.63	22.2	4.07
Volleyball	37.02	5.27	35.85	4.5	29.23	5.03	22.4	3.52
Wrestling	39.25	6.37	36	5.36	29.21	4.29	23.25	2.91
Yoga	37.47	4.4	35.41	2.79	29.41	4.14	24.76	2.56
Untrained	35.59	5.98	34.43	5.39	27.99	4.85	22.16	4.28
Professional athletes	37.9	5.5	36.3	4.6	29.83	4.5	22.88	3.5
**Women**								
**Sport**	**Perception of Emotion**		**Managing Own Emotion**		**Managing Others’ Emotion**		**Utilization Emotion**	
	**Mean**	**SD**	**Mean**	**SD**	**Mean**	**SD**	**Mean**	**SD**
Aerobics	38.55	35.87	30.92	23.73	3.53	4.36	4.65	5.62
Basketball	38.53	36.28	31.03	23.34	3.78	4.21	4.59	5.61
Callanetics	37.06	35.71	30.74	23.23	3.91	4.04	5.01	5.66
CrossFit	36.47	34.92	29.26	22.92	3.56	4.63	4.44	5.54
Cycling	39.19	36.73	31.06	23.31	3.82	4.35	4.32	5.58
Dance	39.35	37.02	31.81	24.36	3.50	4.35	4.61	5.75
Fitness	38.79	36.35	30.90	23.94	3.82	4.35	4.89	5.73
Gym	38.02	36.22	30.64	24.19	4.22	4.88	5.74	7.28
HIT	38.00	36.47	30.40	23.87	2.72	3.89	4.37	6.43
Independent exercise	38.08	35.76	30.46	23.60	4.02	4.23	5.22	5.84
Martial arts	38.63	36.38	30.29	24.06	3.02	3.79	4.52	6.00
Pilates	39.46	36.90	31.69	24.18	3.62	3.73	4.60	4.91
Resistance exercise	37.67	36.36	30.64	24.13	3.49	4.51	4.76	5.90
Running	38.24	36.07	30.40	23.48	3.87	3.93	4.93	5.74
Stretching	37.37	36.67	30.90	24.20	3.28	4.96	5.68	6.48
Swimming	39.12	36.45	30.70	23.43	3.94	5.64	5.00	6.53
Tennis	38.94	35.81	31.11	23.85	3.50	4.43	4.99	5.35
Track and field	38.55	36.59	30.88	24.20	3.89	4.62	5.16	4.98
Volleyball	37.97	37.33	31.71	24.43	3.97	4.27	5.28	6.28
Yoga	38.65	35.98	30.48	24.00	3.71	4.63	5.14	5.88
Untrained	37.23	34.75	29.85	23.23	3.94	4.70	5.56	6.03
Professional athletes	39.20	36.90	31.00	23.90	3.60	4.10	4.10	5.40

### 3.4. Ranking of the 20 Types of Exercise according to Logical Thinking

The associations between type of exercise and logical thinking (analyzed using one two-way ANOVA) were as follows: women: *p* = 0.108, ${ŋ}_{P}^{2}$ = 0.013; men: *p* = 0.79; ${ŋ}_{P}^{2}$ = 0.15. The associated with of age on LT was not significant both in women (*p* = 0.243) and men (*p* = 0.059) (interaction between factors was not significant). For men and women, LT did not differ significantly between the 20 types of exercise, except that the LT score was higher in men who participate in triathlon than in those who perform boxing and in women who perform CrossFit than in those who participate in HIT exercises (Table 6).

The structures for the solving of logic tasks are shown in Figure 2. The ability to solve the logic tasks was not significantly related to the type of exercise (women: chi-square 109.1, *p* = 0.23; men: chi-square 83.4, *p* = 0.87). The lowest problem-solving score was in the men who participate in boxing; only 8% solved all three tasks (Figure 2). By contrast, all three tasks were solved by 30% of the men who participate in track and field, which was the best performance among men. Among women, higher percentages of those who exercise at a gym/health center (25%) or perform CrossFit (23.7%) solved all three tasks, whereas lower percentages of those who play tennis (11.1%) or participate in HIT exercises (13.3%) solved all three tasks.

Figure 3 shows the ratings of the 20 most popular exercise types together with EI and LT scores. Groups within the top five EI and LT scores were women who participate in cycling and men who perform martial arts or track and field.

### 3.5. Relationships between Indicators

We found negative (but not significant) correlations between EI and LT for men and women: r = –0.37 (*p* = 0.107) and r = –0.39 (*p* = 0.09), respectively (the correlation coefficient among EI and LT when men and women together = –0.62 (*p* < 0.001)). LT did not correlate significantly with sleep in women (r = 0.48; *p* > 0.05) but did in men (r = 0.67; *p* < 0.05). MVPA and EI correlated significantly in women (r = 0.66) and men (r = 0.77), respectively (*p* < 0.05 for both).

## 4. Discussion

To our knowledge, this the first study to report that any exercise (professional sport, independent exercise, or exercise at a gym/health center) is significantly related to EI in men and women. We found that female professional athletes solved the lowest number of logic tasks compared with women in other groups, although the LT was better in women who exercise independently or at a gym/health center than in those who do not exercise. Among men, LT was not related to the type of exercise. The unique aspects of our study are the ranking of the 20 most popular types of exercise in Lithuania according to EI and logical thinking (LT) and the analysis of the relationships between exercise type and EI and LT in men and women separately. We found a negative (but not significant) correlation between EI and LT in both men and women, a positive correlation between MVPA and EI in both men and women, and no significant correlation between MVPA and LT (the correlation coefficient was calculated between the top “20”). Interestingly, in the “20” exercises, in women and men, neither EI nor LT depended on exercise specifics, but EI for both women and men depended on age.

This is consistent with data from other studies that also reported direct associations between EI and exercise [3,29,30,31,32,33]. It is not surprising that people who exercise have better EI because they strive to be the best and often want to lead, and EI is an important trait for leadership [34]. In our study, EI scores were not higher in professional athletes than in nonprofessionals who exercise independently or at a gym/health center. One recent systematic review showed that EI is a determining factor in the improvement in sports competencies [31]. Stanković et al. concluded that elite athletes are characterized by positive, high scores in self-efficacy and emotionality [32]. Ubago-Jiménez found that the relationship between PA and EI is stronger in male than in female students [33]. Our finding of significant positive correlations between MVPA and EI for both men and women is consistent with earlier work. Other studies suggested that EI is related to PA [3] and rapid decision making [35].

Few studies have examined whether EI is related to type of exercise. Stanković et al. showed that professional judo athletes are characterized by a low degree of emotionality, but the emotionality of members of team sports is higher [32]. In our study, EI was highest among participants in dual sports (martial arts, wrestling, boxing) and lowest among those who perform powerlifting. Mills et al. found that emotional competence is an important trait in the training of talented football players [29]. We found that EI of women was highest among those who dance and lowest among those who participate in CrossFit.

We found lower LT in professional athletes than in other groups that exercise, and higher LT in physically active women compared with those do not exercise. However, in men, LT was not related to the type of exercise. Several studies are currently looking at associations between PA and cognitive functions [4,5,8,12,16,36,37,38] or learning achievements [39,40,41,42]. To determine the effectiveness of intuitive thinking and LT, we selected the CRT [26]. CRT is used widely in the study of cognitive ability (intuitive thinking and LT) [43,44]. For example, women who participate in CrossFit had the highest LT and the lowest EI, and men who participate in triathlon had the highest logic score, and their EI was one of the lowest. In addition, the EI of men who participate in boxing was one of the highest, whereas their LT score was among the lowest.

There is a lack of research to explain which types of exercise or PA best stimulate LT (if at all). An increasing number of studies have investigated changes in the functional and structural organization of brain networks in response to exercise and increased cardiorespiratory fitness (CRF) [37]. Higher PA and CRF levels are associated with better cognitive and motor function, which may reflect enhanced structural network integrity. Dimech et al. reported sex differences in the associations between fitness and brain function in older adults [45]. Ferreira et al. found that, although various physical exercises (aerobic, neuromuscular, flexibility, or neuromotor) stimulate brain activity in older adults, balance and coordination exercises (neuromotor) are best performed [38]. Working memory in children and adolescents is trained better with high coordinative demands [46]. However, in our study, the LT of men was the highest among those who participate in triathlon, a sport that does not require much coordination, balance, or motor control.

We found no significant association between MVPA and LT in men or women, but LT was significantly associated with sleep duration in men. For example, women who cycle were in the top five for EI and LT, and men who participate in martial arts or track and field were equally good in both LT and EI. This is consistent with the findings of Giordano et al.: that children who practice martial arts have better executive functioning and higher school marks than those involved in team sports or not involved in any sports [47]. One study concluded that judo and ball games are beneficial to the quality of sleep and life of children and adolescents [48]. Another study reported that walking and bicycling rated much higher in measures of physical and mental health, confidence, positive affect, and overall hedonic well-being, which suggests significant benefits of physically active commuting [49]. By contrast, watching fast-paced movies impairs children’s executive function [50], and greater media multitasking is associated with poorer executive function ability, worse academic achievement, and a lower growth mind-set [51]. However, despite the data noted above, it is not yet possible to conclude which types of exercise are best for EI or LT or which may affect both LT and EI.

## 5. Limitations and Directions for Future Research

The main limitation of our research is that we could not test the development of EI and LT over several years experimentally in people participating in 20 of the most popular types of exercise. We also cannot rule out that, in addition to the type of exercise, it is possible that the participants with higher EI and LT choose the type of exercise most appropriate for them. In addition, the participants of our top “20” exercises differed by age. A person’s choice of PA is influenced by a number of interrelated determinants such as demographics, health, and health behavior and psychological, social, and environmental determinants related to the intervention [52]. We can only speculate that people make the decision to be physically active and select a particular form of exercise based more on implicit knowledge than on explicit (reflective) thinking because implicit thinking requires less effort and is more psychologically attractive [21]. To standardize the prescription of exercise for older adults, the American College of Sports Medicine (ACSM, 2017) has classified physical exercise into four types: aerobic, neuromotor, neuromuscular, and flexibility; however, in our study, the 20 popular types of exercise did not fit neatly into these categories [53].

## 6. Conclusions

In summary, any exercise (professional sport, independent exercise, or exercise at a gym/health center) has a significant association with on EI in men and women. However, women in professional sports solved the lowest number of logic tasks (LT) compared with women in other groups, although the LT was better in nonprofessional women who exercise independently or at a gym/health center than in those who do not exercise. Among men, logical thinking was not related to the type of exercise. We found only the tendency for a negative correlation between EI and LT in the 20 most popular types of exercise. and We found a positive correlation between MVPA and EI. The unique aspects of our research are the ranking of 20 of the most popular types of exercise in Lithuania and the analysis of the relationships between exercise type and EI and LT for men and women separately. The long-term associated with of these popular exercise types on EI and LT needs to be confirmed and tested experimentally.

## Figures and Tables

**Figure 1 ijerph-19-10076-f001:**
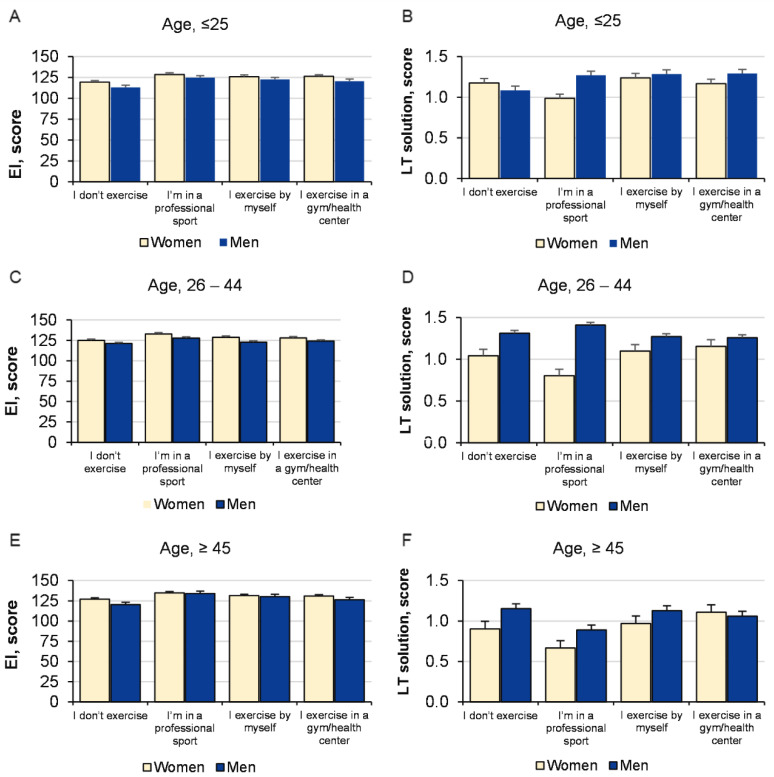
Emotional intelligence (**A**,**C**,**E**) and the effectiveness of solving different logic tasks (**B**,**D**,**F**) among men and women of different ages (years) who do not exercise, are professional athletes, exercise independently, or exercise at a gym/health center.

**Figure 2 ijerph-19-10076-f002:**
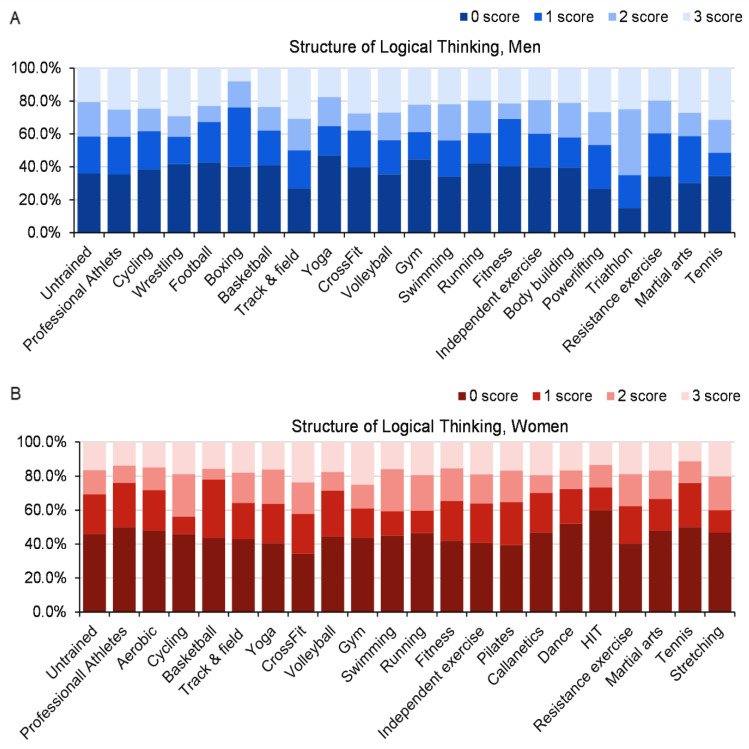
Ability to solve three logical thinking tasks among men (**A**) and women (**B**) according to the type of exercise. The numbers at the bottom (0–3) indicate the number of tasks solved.

**Figure 3 ijerph-19-10076-f003:**
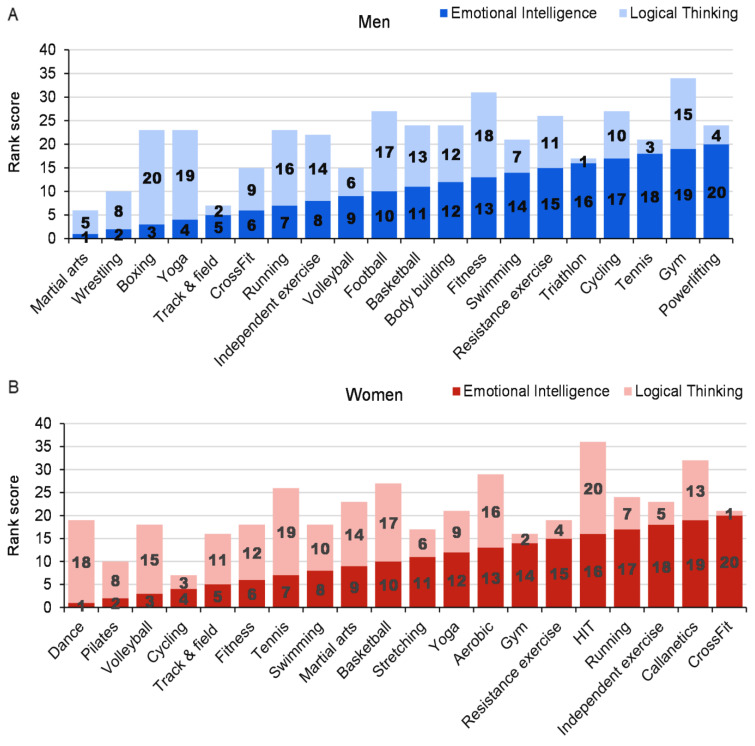
General ranking according to emotional intelligence and logical thinking men (**A**) and women (**B**).

**Table 3 ijerph-19-10076-t003:** Descriptive characteristics, body mass index, sleep, and moderate-to-vigorous physical activity in men and women who exercise, listed according to the type of exercise.

			**Age, Years**		**Body Mass Index, kg/m^2^**		**Experience, Years**		**Sleep, METs**		**Moderate-to-Vigorous PA, METs**	
**Men**	**N**	**N Second** **Sport**	**Mean**	**SD**	**Mean**	**SD**	**Mean**	**SD**	**Mean**	**SD**	**Mean**	**SD**
Basketball	161	20	32.61	9.46	25.39	3.26	17.83	9.98	6.52	0.8	19.3	12.3
Running	152	26	37.26	10.21	25.08	3.05	9.23	8.72	6.49	1	18.22	10.72
Resistance exercise	126	27	32.98	9.5	26.33	3.3	7.63	7.95	6.44	0.78	15.68	12.27
Independent exercise	113	10	38.93	13.37	26.07	4.42	8.67	11.19	6.44	0.97	18.72	14.12
Cycling	73	28	40.62	10.33	24.85	3.31	9.59	10.38	6.42	0.8	19.48	10.7
Martial arts	70	1	37.1	12.38	25.47	3.6	12.79	10.86	6.36	0.92	21.21	11.61
Football	61	13	32.38	10.23	25.18	3.18	18	9.92	6.6	1.14	23.1	13.62
CrossFit	58	5	34.69	7.6	26.4	2.61	3.79	5.66	6.69	0.76	15.58	9.43
Volleyball	48	19	33.27	10.09	25.46	3	10.49	10.29	6.73	0.84	20.03	9.47
Fitness	42	10	35.55	10.65	26.25	3.62	10.37	10.43	6.56	0.78	19.35	12.11
Swimming	41	17	39	1384	25.79	3.47	13.98	18.86	6.8	0.69	17.89	13.38
Bodybuilding	38		32.71	8.53	27.53	4.02	7.54	5.81	6.42	0.93	16.15	13.7
Tennis	35	10	39.94	12.96	25.27	2.13	7.6	7	6.56	0.75	19.73	10.94
Track and field	26	4	32.54	11.85	24.57	3.2	12.31	8.95	6.99	1.03	19.62	11.21
Boxing	25	6	31.72	7.98	25.47	2.72	10.39	7.4	6.44	0.67	20.13	10.55
Wrestling	24	2	33.13	9.67	27.3	3.85	14.83	11.26	6.75	0.81	18.95	9.53
Gym	22	9	32.94	6.83	26.28	3.42	8.29	6.86	6.6	0.69	14.44	15.26
Yoga	21	7	44.29	11.34	26	2.74	4.97	4.55	6.72	0.71	12.71	9.75
Triathlon	20	1	34.65	6.04	24.35	1.92	3.72	2.05	6.84	0.54	17.16	8.19
Powerlifting	20	3	27.67	11.89	28.69	4	5.97	5.52	7.02	0.58	17.22	10.28
			**Age, Years**		**Body Mass Index, kg/m^2^**		**Experience, Years**		**Sleep, METs**		**Moderate-to-Vigorous PA, METs**	
**Women**	**N**	**N Second Sport**	**Mean**	**SD**	**Mean**	**SD**	**Mean**	**SD**	**Mean**	**SD**	**Mean**	**SD**
Independent exercise	564	46	41.55	12.99	24.19	4.40	5.32	7.27	6.57	0.85	13.74	10.37
Yoga	242	56	41.26	11.80	22.86	3.59	5.48	5.83	6.64	0.84	13.98	9.29
Resistance exercise	181	56	35.19	8.70	23.02	3.38	3.96	4.55	6.65	0.82	14.88	9.28
Running	176	51	36.47	10.35	22.46	2.98	7.81	8.46	6.57	0.80	17.80	9.81
Pilates	162	32	43.13	9.72	23.57	3.53	4.27	5.06	6.60	0.79	12.71	7.85
Aerobics	142	21	37.18	10.37	23.69	3.79	7.77	9.00	6.57	0.82	15.60	9.95
Fitness	136	21	35.64	10.16	22.66	3.06	5.56	6.34	6.53	0.81	15.05	10.54
Dance	127	32	32.85	10.99	23.07	3.86	7.76	9.10	6.73	0.75	16.73	11.14
Callanetics	77	30	41.92	9.90	24.45	3.53	3.61	4.75	6.49	0.74	11.86	7.99
Swimming	69	25	39.67	15.97	24.68	4.75	11.76	13.52	6.67	0.95	13.46	9.66
Gym	64	8	39.63	8.83	23.69	3.85	5.35	6.95	6.49	0.73	16.70	10.18
Volleyball	63	10	33.38	12.51	22.98	2.82	11.20	9.87	6.51	1.08	20.59	12.06
Track and field	56	4	30.04	12.60	22.05	3.66	10.50	9.20	6.99	0.81	22.30	11.88
Tennis	54	4	35.39	10.40	23.16	3.41	9.52	8.23	6.47	1.01	18.02	9.19
Cycling	48	24	41.10	10.73	25.08	5.68	8.47	9.41	6.58	0.88	16.13	11.52
Martial arts	48	1	36.69	14.18	22.81	2.56	6.99	6.91	6.28	0.78	16.16	9.12
CrossFit	38	6	32.13	6.96	23.04	2.87	2.29	1.65	6.78	0.79	16.41	9.39
Basketball	32	2	32.28	11.33	24.23	5.26	19.03	10.66	6.54	0.96	19.18	12.30
Stretching	30		43.60	10.12	24.62	4.48	5.79	6.50	6.65	0.77	12.96	11.47
Cardio	15	15	36.73	5.84	22.95	2.33	6.17	6.83	6.27	0.94	16.65	13.40

**Table 6 ijerph-19-10076-t006:** Relationship between logical thinking and type of exercise in men and women.

Men			Women		
Sport	Logic		Sport	Logic	
	Mean	SD		Mean	SD
Triathlon	1.75	1.02	CrossFit	1.32	1.19
Track and field	1.54	1.21	Gym	1.20	1.25
Tennis	1.49	1.27	Cycling	1.17	1.21
Powerlifting	1.47	1.19	Resistance exercise	1.16	1.15
Martial arts	1.39	1.18	Independent exercise	1.14	1.15
Volleyball	1.35	1.23	Stretching	1.13	1.22
Swimming	1.32	1.17	Running	1.13	1.20
Wrestling	1.29	1.3	Pilates	1.12	1.11
CrossFit	1.26	1.25	Yoga	1.12	1.11
Cycling	1.25	1.21	Swimming	1.12	1.16
Resistance exercise	1.25	1.13	Track and field	1.11	1.15
Bodybuilding	1.24	1.2	Fitness	1.08	1.11
Basketball	1.2	1.21	Callanetics	1.03	1.17
Independent exercise	1.19	1.16	Martial arts	1.02	1.16
Gym	1.17	1.25	Volleyball	1.02	1.13
Running	1.17	1.18	Aerobic	0.95	1.10
Football	1.13	1.2	Basketball	0.94	1.08
Fitness	1.12	1.17	Dance	0.92	1.14
Yoga	1.06	1.2	Tennis	0.85	1.04
Boxing	0.92	0.95	HIT	0.80	1.15
Untrained	1.26	1.15	Untrained	1.01	1.12
Professional athletes	1.31	1.2	Professional athletes	0.86	1.00

## Data Availability

The data presented in this study are available on request from the corresponding author.

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
