# Peer review of "What Types of Exercise Are Best for Emotional Intelligence and Logical Thinking?"

_ijerph, 2022, doi:10.3390/ijerph191610076_

Round 1

Reviewer 1 Report

Dear Authors

 Thanks for this really interesting study.

1. Throughout please try to avoid using the word 'effect' to describe the associations between exercise and aspects of cognition - these cross sectional results, as you acknowledge, cannot be used to infer causality.

2. Figures 2-4 could be presented as supplementary tables with numerical values - differences between exercise types seem very small

3. I would like to see some scatter plots of the correlations between LT and EI   separately for men and women

4. I wondered if age mattered?  Did you consider Analysis of covariance, looking at the association between age and LT or IT with exercise type as a fixed effect

5. The difference in male versus female respondents was very high - I wonder why?

6. More could be said about the validity of the EI and LT scales.

Author Response

August 9, 2022

Dear Reviewers,

Thank you very much for the your’s comments concerning our manuscript titled: What types of Exercise are best for Emotional Intelligence and Logical Thinking? Those comments are all valuable and very helpful for revising and improving our paper. We have studied comments carefully and have made correction which we hope will meet with approval. Revised portion are marked in yellow in the paper.

The main corrections in the paper and the responses to the reviewer’s comments are in the attached document.

Reviewer 2 Report

Dear authors: 

I just give you some recommendations: 

First, In the line 14 and 15 of the abstract it is necessary and small introduction of the paper not the number of participants of the study. Please, change this information

Second, in tables and graphics is not necessary to use contractions of the verbs. Please write all the words. 

Author Response

August 9, 2022

Dear Reviewer,

Thank you very much for the your’s comments concerning our manuscript titled: What types of Exercise are best for Emotional Intelligence and Logical Thinking? Those comments are all valuable and very helpful for revising and improving our paper. We have studied comments carefully and have made correction which we hope will meet with approval. Revised portion are marked in yellow in the paper.

The main corrections in the paper and the responses to the reviewer’s comments are as following:

We corrected and supplemented:

Reviewer: 2

First, In the line 14 and 15 of the abstract it is necessary and small introduction of the paper not the number of participants of the study. Please, change this information

We corrected abstract.

Second, in tables and graphics is not necessary to use contractions of the verbs. Please write all the words. 

We corrected tables and graphics.

Round 2

Reviewer 1 Report

Dear Authors,

Thanks for your revisions.

 I cannot recommend acceptance of the paper until you edit the paper to stop using 'effect'.

 A cross-sectional study such as this one is estimating associations between two variables, it is not estimating the effect of one variable on another.

I see you mention using an English Editing service - this is a matter of scientific inference, not of grammar.

Author Response

Dear Reviewer,
Thank you very much for the your’s comments.
1. We change "effect" to "influence" (Revised words are marked in green in the paper).
2. We used two-way ANOVA. Determine the influence and interaction between the, of each factor independently. In our case interaction between age and exercise specific was not significant. 
